# Computational "Accompaniment" of the Introduction of New Mathematical Concepts

**Andrey Lavrenov, Elena Tolkacheva**  **and Sergei Pozdniakov *** 

Department of Algorithmic Mathematics, Saint Petersburg Electrotechnical University "LETI", ul. Professora Popova 5, 197022 St. Petersburg, Russia; avlavrenov@etu.ru (A.L.); eatolkacheva@etu.ru (E.T.)
* Correspondence: pozdnkov@gmail.com

**Abstract:** The computational capabilities of computer tools expand the student's search capabilities. Conducting computational experiments in the classroom is no longer an organizational problem. This raises the "black box" problem, when the student perceives the computational module as a magician's box and loses conceptual control over the computational process. This article analyses the use of various computer tools, both existing and specially created for "key" computational experiments, that aim at revealing the essential aspects of the introduced concepts using specific examples. This article deals with a number of topics of algebra and calculus that are transitional from school to university, and it shows how computational experiments in the form of a "transparent" box can be used.

**Keywords:** computations; mathematics; conceptual understanding; horizontal connections in learning



## 1. Introduction

In the methodology of teaching mathematics in the Soviet Union in the 1950s, 1960s, 1970s, and partly in the 1980s, the formation of mathematical concepts relied heavily on the operational activities of students. This corresponded to the activity approach well developed by Soviet psychologists. The psychological theory of activity was created in Russian psychology due to the works of L. S. Vygotsky, S. L. Rubinshtein, A. N. Leontiev, A. R. Luria, A. V. Zaporozhets, P. Ya. Galperin, and many others. The most complete theory of activity is presented in the works of A. N. Leontiev, in particular in his last book *Activity. Consciousness. Personality* [1]. At the level of mathematics teaching methodology, this was manifested by studying algebra as a separate subject, in which much attention was paid to algebraic calculations.

The psychological basis of this approach to teaching mathematics was studied in detail by I.S. Shapiro and described in the work *From Algorithms to Judgments* [2]. In this work, I.S. Shapiro discusses the operator-logical form of knowledge, which is well consistent with the methodology of teaching school algebra as a subject that studies the transformation of algebraic expressions from one form to another.

The main idea of the book is the "convolution of algorithms", which he considers as a form of generalization and as a mechanism for "running ahead" in solving complex problems.

Let us give an example from this work [2] (p. 231):

"Let us describe in general terms one of the experiments. It took less than three minutes for a math-savvy student A to solve the problem.

Simplify:

$$\frac{2cos^3\alpha - cos\alpha}{2tg\left(\frac{\pi}{4} - \alpha\right) \cdot sin^2\left(\frac{\pi}{4} + \alpha\right) \cdot cos\alpha}$$

We asked the student to tell the way of thinking. Student A wrote on the fly: $2cos^3\alpha - cos\alpha = cos\alpha \cdot cos2\alpha$

Experimenter:—Do you know such a formula?

A: It's easy.

Experimenter: Do you remember this formula?

A:—I will derive this formula. (Makes, without hesitation, the necessary transformations.)

Experimenter:—Did you solve it in your head?

A:—So in detail—no, I immediately saw what was happening.

Experimenter:—But did you somehow deduce when you solved?

A: It seems not. If you do everything, then one thing will "overwhelm" the other ... You have to think about what is not obvious ...

Experimenter:—For what reasons did you replace $tg\left(\frac{\pi}{4} - \alpha\right) = ctg\left(\frac{\pi}{4} + \alpha\right)$?

A:—I noticed that it turns out:

$$2ctg\left(\frac{\pi}{4} + \alpha\right) \cdot sin^2\left(\frac{\pi}{4} + \alpha\right) = 2cos\left(\frac{\pi}{4} + \alpha\right)sin\left(\frac{\pi}{4} + \alpha\right) = sin\left(\frac{\pi}{2} + 2\alpha\right) = cos2\alpha$$

Experimenter:—You reasoned in such detail?

A:—No, I immediately saw that it turns out $cos2\alpha$, etc."

Approximately the same way of "thinking aloud" was observed when the problem was solved by other students gifted in mathematics. The presence of a folded system of inferences ensured the simultaneous and quick consideration of several actions and the choice of a way to solve a task. Of the eight ninth-graders gifted in mathematics who participated in the experiment, six solved the problem orally, and two with a minimum number of records of intermediate equalities.

"Running ahead" suggests that the actions to transform trigonometric expressions not only pass into the internal plane and turn into thought processes, but also act as objects that the student operates on, building a plan for solving the task. These processes are integrally considered in the APOS theory, the main ideas of which will be outlined below.

The emergence of powerful mathematical tools for performing symbolic calculations, such as Maxima, Mathematica, Maple, Sage, and MathPartner, has significantly reduced the value of manual calculations, on which the technique of moving from algorithms to judgments through algorithm convolution was based. Computer programs perform calculations faster and without errors. Moreover, programs such as UML (Universal Mathematical Solver) are specifically oriented towards solving school problems in algebra and present not only the answer, but also the chain of transformations that the teacher requires.

Under these conditions, the following questions arise.

- To what extent should the performance of calculations by hand be preserved in the teaching of mathematics at school?
- How to preserve and develop the mathematical culture of students without relying on traditional operational activities?

If it is difficult to answer the first question, then we will try to answer the second question constructively, analyzing various examples of computational schemes related to the formation of mathematical concepts.

## 2. Activity—Interiorization—Encapsulation

Shapiro's idea of the convolution of algorithms fits well with the idea of internalization of external actions. The convolution of algorithms can be considered as one of the manifestations of the psychological mechanism of internalization, that is, the transfer of actions with objects of the external environment to the internal—mental—plan. Another important psychological phenomenon associated with interiorization is "encapsulation", which plays an important role in the APOS (Actions, Processes, Objects and organizing them in Schemas) theory [3] "... to describe how actions become interiorized into processes and then encapsulated as mental objects, which take their place in more sophisticated cognitive schemas..." [4].

This statement can be explained as follows: the performing of an action with objects of the external environment (actions) transfers them through the process of internalization

into mental processes (processes), which in turn are folded (encapsulated) into objects (objects), on which mental activity (schemas) is built.

At the same time, if Shapiro associates the performance of calculations exclusively by hand, Dubinsky considers the possibility of replacing calculations by hand with digital symbolic calculations.

Of interest is the existence of the concept of "encapsulation" in a different sense, as one of the main components of object-oriented programming.

John D. Cook believes that the use of the same term in programming and psychology is not accidental; he calls encapsulation in programming "logical", and the phenomenon of encapsulation as a mechanism of thinking "psychological encapsulation" [5]:

"A piece of software is said to be encapsulated if someone can use it without knowing its inner workings. The software is a sort of black box. It has a well-defined interface to the outside world. "You give me input like this and I'll produce output like that. Never mind how I do it. You don't need to know".

I think software development focuses too much on logical encapsulation. Code is logically encapsulated if, in theory, there is no logical necessity to look inside the black box.

. . .

Maybe there's nothing wrong with the code, but you don't trust it. In that case, the code is logically encapsulated but not psychologically encapsulated. That lack of trust negates the psychological benefits of encapsulation... A failure of logical encapsulation is objective and may easily be fixed. A loss of confidence may be much harder to repair".

Thus, if the object is psychologically encapsulated, then the student is fluent in it, using it to produce complex judgments. At the same time, a logically encapsulated object can exist in a program as a "black box" that one can work with, but it is not an element of human mental activity. The challenge for methodists is to make logical and psychological encapsulation become part of a single whole. One of the ways is to de-encapsulate the object represented by logical encapsulation so that by working with it in an "expanded" form through internalization, one could achieve psychological internalization. It should be noted that formally assimilated definitions of mathematical concepts can also be classified as logically encapsulated objects. With the formal assimilation of mathematical knowledge, as written by mathematician A.Ya. Khinchin [6], the student cannot use knowledge, provide examples, solve problems, although he/she can correctly pronounce the formulations of definitions and theorems:

"Those who have taken out of school only external, formal expressions of mathematical methods, without having mastered their substantial essence, when they meet a real problem, will, of course, be deprived of the opportunity to see which of these methods can be applied to its solution. He will not be able, as we say, to formulate a practical problem mathematically; to a large extent, he will be helpless in solving this problem, since he has not developed the habit of really comprehending the formal operations performed, as a result of which neither the interests of the practical task facing him, nor even the mathematical content of the emerging problems will be able to guide him when choosing these operations" [6] (pp. 21–27) (translated by the authors of the article).

Next, we will consider examples of the introduction of mathematical concepts that demonstrate how computational processes can be used to de-encapsulate concepts given by verbal definitions, and introduce new concepts based on the analysis of the computational scheme.

### 3. Positional Numeral Systems and Information Compression Algorithms

In grade school, school students begin to add numbers in the unary numeral system, when the value of a number is determined as the cardinality of a set of sticks or matches. The addition algorithm in this system is extremely simple—the students need to connect together two piles corresponding to the terms. Later, at university, they will return to the unary numeral system when they study the theory of algorithms and build Turing machines. Then, the task of constructing an algorithm can be described, for example, as

III + II → IIIII. Further, in grade school, students are introduced to the decimal numeral system (and in computer science classes in high school also to the binary numeral system) and students study addition algorithms in these numeral systems. Between the introduction of the concept of a number through the unary number system and the further study of algorithms for numbers in the decimal system, a logical gap arises—why was it necessary to introduce a positional numeral system, if it is much more difficult to add numbers in it than in the unary one? To answer the question, the following calculations can be made. Let us calculate how much ink we need to spend on writing numbers in different numeral systems. We will assume that one drop is needed to write a one, and three drops to write a zero. Then, the number "ten" in the unary system will be written as IIIIIIIIII and will require ten drops, and in decimal it will be written as 10 and will require 1 + 3 = 4 drops. If we need to write the number "one hundred" in different systems, then in the unary system, ninety more sticks will need to be added to the number ten, which will require ninety drops of ink, whereas only three additional drops will be required to go from "ten" to "one hundred" in decimal. To go from "one hundred" to "one thousand" in a unary numeral system, nine hundred drops are required, whereas in decimal only another three drops.

Thus, the purpose of the transition to the decimal number system is to compress information. To give students an even better idea of what compression means, we can offer to calculate the length of the string that represents the number "one thousand" in different number systems. If one centimeter is allocated to one digit, then to write the number "one thousand" in the unary numeral system, we will need to write a line ten meters long, and to write the number "one million" a line a thousand times longer, that is, ten kilometers long, while in the decimal system, "one million" will be seven centimeters long.

Thus, simple calculations show that it is reasonable to introduce a decimal notation in order to more compactly encode information—information compression—a concept that is important for computer science. From this point of view, the transition to a new base by division can be considered as an information compression algorithm. The actions that are performed in this case are carried out in elementary school: arrange the sticks (matches) into piles of ten each, then do the same with these piles, arranging the piles into groups of ten each, etc. Formally, this algorithm can be described as follows, where the mod and div operations should be considered as operations with heaps (in this interpretation, they are carried out by one operation connecting mod and div), described above, and output—as fixing the next "digit".

**Computational algorithm**
*while* N ≠ 0
*output (N mod 10);*
*N : = N div 10*
*end while*

*3.1. P-Adic Numbers and the Algorithm "Division with Remainder"*

An amazing example of how the essence of a mathematical concept can be expressed through calculations is *p*-adic numbers. Here is a standard definition from the mathematical literature, which even mathematically gifted students cannot immediately understand.

**Definition 1.** *An integer p-adic number for a given prime p is an infinite sequence* $a = \{a_1, a_2, ....\}$ *of residues* $a_n$ *modulo* $p^n$ *satisfying the condition:* $a_n \equiv a_{n+1} (mod\, p^n))$ [7].

Consider the computational process of the algorithm for converting a natural number N into a positional system with base $p \geq 2$:
**Computational algorithm**
*k :=0;*
*while (N ≠ 0)*
$a_k$ := N mod p;
*N := N div p;*

*k := k + 1*
***end while***

Let us apply this algorithm to the "forbidden"—negative—number, for example, to $N = -1$. Let us take as an example the smallest prime number $p = 2$.

The first step of this algorithm gives 1 as the remainder ($a_0 = 1$) and $-1$ as the quotient ($N = -1$).

The algorithm loops and the output is an infinite sequence of ones: $(\ldots 111) = (\ldots a_2\, a_1\, a_0)$.

Consider another computational process defined by the algorithm for adding numbers in the positional number system:

**Computational algorithm**
*k := 0; s := 0;*
***do***
$c_k := a_k + b_k + s \bmod p;$
$s := a_k + b_k + s \; div \; p$
***until*** $(a_k = 0 \; u \; b_k = 0)$
***end do***

It is unusual that this algorithm, like the previous one, does not stop if one of the terms is given by an infinite sequence.

For example, if we add $-1$, written as a sequence (...111) with the number 4, written in binary, we get:

...1 1 1 1 1
<u>    1 0 0</u>
...0 0 0 1 1

that is, the number 6 in binary notation (if we do not take into account the infinite number of zeros that precede the first unit from the left).

After these calculations, one could "come up with" another definition, for example, the one which is given in Wikipedia:

"In number theory, given a prime number $p$, the $p$-adic numbers form an extension of the rational numbers which is distinct from the real numbers, though with some similar properties; $p$-adic numbers can be written in a form similar to (possibly infinite) decimals, but with digits based on a prime number $p$ rather than ten, and extending (possibly infinitely) to the left rather than to the right. Formally, given a prime number $p$, a $p$-adic number can be defined as a series

$$s = \sum_{i=k}^{\infty} a_i p^i$$

where $k$ is an integer (possibly negative), and each $a_i$ is a integer such $0 \le a_i < p$. A $p$-adic integer is a $p$-adic number such that $k \ge 0$".

It should be noted that even if the last definition is given before the operation of the two algorithms above is shown, understanding the concept of a $p$-adic number will present significant difficulties.

It is important to note that the idea of $p$-adic numbers is used at the "lower level" of computer calculations: the inverse binary code of integers is nothing but a 2-adic representation with a fixed number of digits.

It can be concluded that some mathematical concepts are comprehended through computational algorithms.

*3.2. Diophantine Equations, Continued Fractions, and Euclidean Algorithm for Finding GCD*

In the course of algebra and/or in the course of discrete mathematics at technical universities, such concepts as the greatest common divisor, Bézout's identity, continued fractions, convergents, linear Diophantine equations, and modular reciprocal are studied. Usually, the introduction of these concepts is accompanied by verbal definitions, from which the connection of these concepts with certain algorithms is not visible. At the same time, all of the above topics are united by the Euclidean algorithm. However, when presenting the material, this fact fades into the background, while the general computational scheme

can be used as a tool for forming a general idea that connects these concepts. Moreover, it can be used to derive an algorithm for constructing convergents from the GCD linear representation algorithm (extended Euclidean algorithm).

We have created a special environment, which is based on the table representing a computational process that combines the calculation of quotients, remainders, and linear representations of remainders through the original pair of numbers.

The special environment implements the following algorithm:

**Computational algorithm**

$(x_a;y_a) := (1;0);$ $(x_b;y_b) := (0;1);$
*while* $a \neq 0$ *and* $b \neq 0$
$q := a \ div \ b;$
$r := a - q \cdot b;$ $(x_r;y_r) := (x_a;y_a) - q \cdot (x_b;y_b);$
$a := b;$ $(x_a;y_a) := (x_b;y_b);$
$b := r;$ $(x_b;y_b) := (x_r;y_r)$
*end while*

In this algorithm, a and b are the original numbers, and $q$ and $r$ are the quotient and the remainder when $a$ is divided by $b$. In the algorithm, after each iteration of the loop, the variables a and b are assigned the values $b$ and $r$, respectively.

The vectors $(x_a;y_a)$, $(x_b;y_b)$, $(x_r;y_r)$ are vector representations of the numbers $a$, $b$, and $r$. With them in the algorithm, the same actions are performed as with the numbers $a$, $b$, and $r$.

Thus, the presented algorithm combines the regular and extended Euclidean algorithms.

If in this algorithm we replace subtraction with addition and change the initialization of vectors in the first line of the algorithm, we get an algorithm for constructing a continued fraction for a rational number $a/b$ and convergents for this continued fraction.

**Computational algorithm**

$(x_a;y_a) := (0;1);$ $(x_b;y_b) := (1;0);$
*while* $a \neq 0$ *and* $b \neq 0$
$q := a \ div \ b;$
$r := a + q \cdot b;$ $(x_r;y_r) := (x_a;y_a) + q \cdot (x_b;y_b);$
$a := b;$ $(x_a;y_a) := (x_b;y_b);$
$b := r;$ $(x_b;y_b) := (x_r;y_r)$
*end while*

It is easy to prove that both algorithms will generate numbers of the same absolute value. For coprime numbers a and b, the last pair of numbers in the extended Euclidean algorithm will be (b;-a); that is, in absolute value, it will give the original pair of numbers in the reverse order. This consideration makes it possible to explain the transfer of the extended Euclidean algorithm to the algorithm for constructing convergents.

Thus, understanding the work of similar computational algorithms leads to the realization of more general ideas underlying them, the connection of different representations of these general concepts, and the transfer from the algorithm to the proof of theorems. The latter suggests that computations can become the basis for both the psychological and logical encapsulation of a new concept.

The other side of the analyzed example is the methodological aspect associated with the creation of this environment based on the existing logical connection between the various topics of the mathematics course.

As can be seen from Figures 1 and 2, the same simple computational base can combine several tasks that are different in subject matter, but close in meaning and algorithms used.

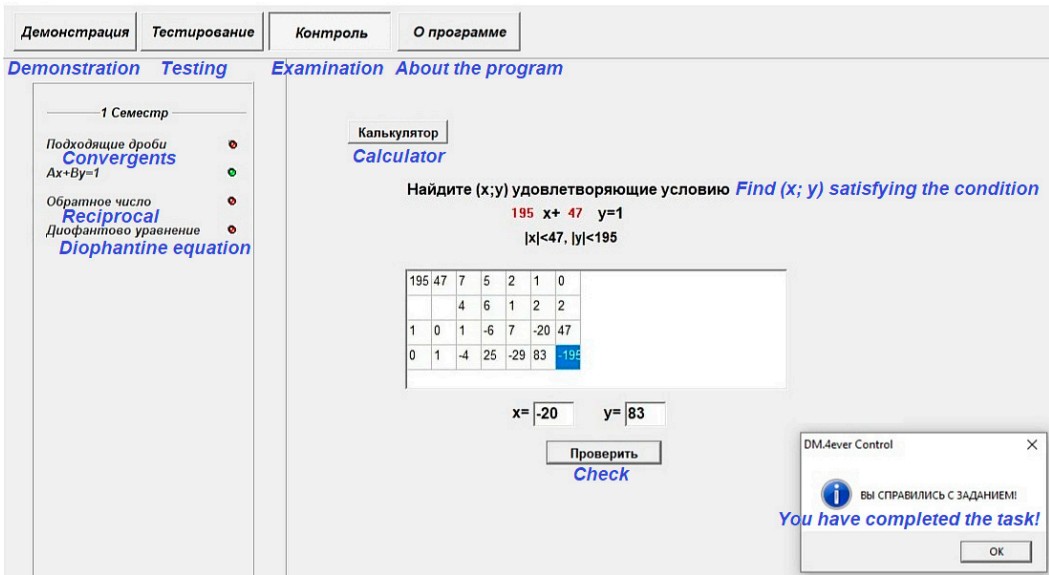

**Figure 1.** Working with the Euclidean algorithm and the extended Euclidean algorithm in the special environment.

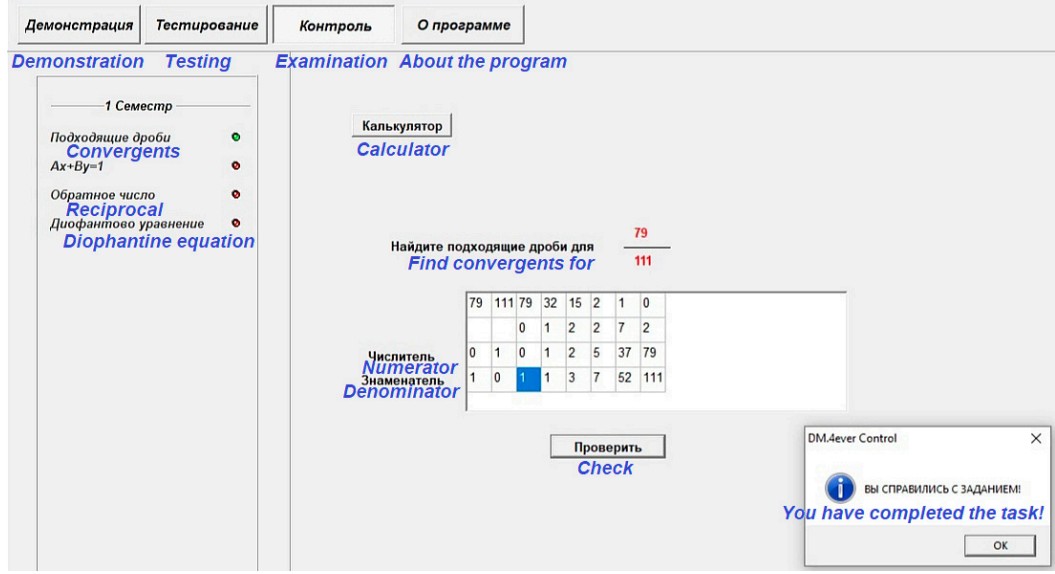

**Figure 2.** Working on the construction of the continued fraction and its convergents in the special environment.

A feature of this module is that the algorithms underlying it are known to the students, and they can not only solve problems, but also study solutions of similar tasks by choosing the demonstration mode. In this mode, the program generates random numbers a and b, checking that the calculation table is neither too large nor too small. In the testing mode, the program will check each move (filling one cell) and highlight the result in green or red color, depending on whether the correct number is entered in the cell or not. Finally, in exam mode, the student completes the entire spreadsheet and it is sent to the server for review. It should be noted that the possibility of opening two programs, one of which works in demo mode and solves the example required for answering the exam, is blocked by the fact that entering numbers into this program is not allowed—tasks are generated automatically.

*3.3. Exponential Functions and Euler's Computational Scheme*

When introducing an exponential function, there is one methodological problem associated with the fact that the exponentiation is introduced in the algebraic paradigm, while the exponential function requires an appeal to the ideas of mathematical analysis. It is necessary to justify the selection of one of the bases—the number *e*—among others. Some textbooks suggest, for example, choosing an $a^x$ function whose slope of the tangent to the graph at zero is equal to 1. It is clear that the appearance of the tangent is associated with the introduction of a derivative, that is, an appeal to the concepts of calculus.

Therefore, it is of interest to consider another process of introducing the concept of an exponent based on differential calculus. But then it will be necessary to combine it with the algebraic ideas on which the introduction of the exponential function is based.

In order to establish a connection with the function $y = a^x$, we recall the steps for introducing this concept. The exponential function is first defined for the natural argument $y = a^n$, and then it is redefined first for negative values of the argument as $a^{-n} = 1/a^n$, and then for rational values of the argument through radicals as $a^{\frac{p}{q}} = \sqrt[q]{a^p}$. The last and the most difficult stage, the definition of an exponential function of a real argument, is also associated with calculus and requires the concept of continuity. This stage is omitted from the pre-college mathematics curriculum.

Let us consider the approach to the introduction of the exponent based on the primacy of the ideas of calculus.

The most natural definition of the function $y = exp(x) = e^x$ is given through the solution of the differential equation $y'(x) = y(x), y(0) = 1$.

Consider the Euler algorithm for the approximate solution of differential equations, which is obtained from the definition of the derivative, if we discard the limit from it: instead of $y'(x) = \lim_{h \to 0} \frac{y(x+h)-y(x)}{h}$ consider $y'(x) = \frac{y(x+h)-y(x)}{h}$, from which we express the value of the function at the next point $y(x + h) = y(x) + y'(x) \cdot h$.

Given the equality $y'(x) = y(x)$, we get $y(x + h) = y(x) + y(x) \cdot h = y(x) \cdot (1 + h)$.

The algorithm for calculating approximate values of exponential function using the Euler method is as follows:

**Computational algorithm**
x := 0; y(0) := 1;
*repeat*
*y(x + h) := y(x)·(1 + h);*
*x := x + h*
*end repeat*

For h = 1, we get the definition of a geometric sequence y(x + 1) := y(x)· 2 under the condition y(0) = 1:

$1, 2, 2^2, 2^3, \ldots$

Thus, the computational process associates a differential equation with a $y = a^x$ function.

Using another computational process, one can show how the number e arises from the scheme of Euler's method.

Consider *h* = 0.1 and express $y(1)$ in terms of $y(0) = 1$:

**Computational algorithm**
*x := 0; y(0) := 1; h = 0.1;*
*repeat* 10 *times*
*y(x + h) := y(x)· 1.1;*
*x := x + h*
*end repeat*

Multiplying 1 ten times by 1.1, we get that y(1) = 1.110 = (1 +1/10)10 ≈ 2.59...

Further, by analogy, if we take the step h = 0.01, then y(1) = 1.01100 = (1 + 1/100)100 ≈ 2.70...

After that, it will be natural to define the natural base as the limit $\lim_{n \to \infty} (1 + 1/n)^n$.

Thus, in this example, the computational scheme for solving the differential equation by the Euler method served as a link between the differential equation and the definition of the exponential function as a generalization of the multiplication operation.

### 3.4. The Concept of the Integral and the Approximate Calculation of the Derivative

Continuing the previous case study, we will show that Euler's computational scheme allows us to connect different representations of the concept of an integral and provide the formation of ideas about the connection between an integral and an antiderivative.

Let us consider the simplest algorithm for approximate calculation of the area of the region under a curve above the interval [0; 1] (Figure 3).

**Computational algorithm**

*S := 0; x := 0;*
*while x < 1*
*S := S + h· f(x);*
*x := x + h*
***end while***

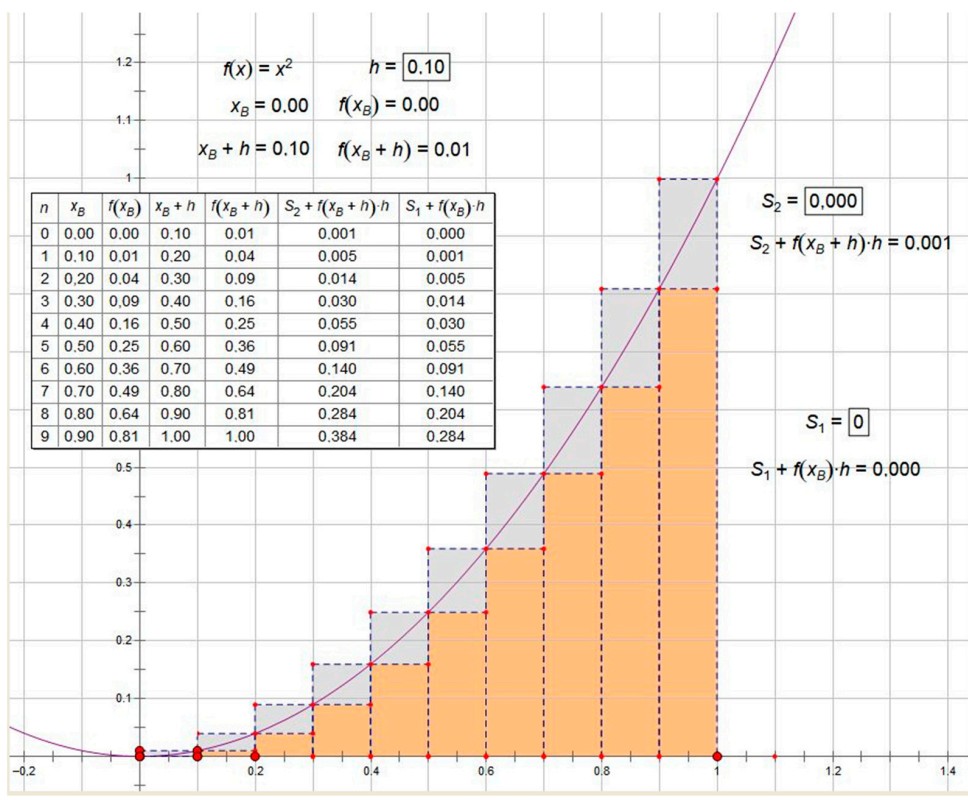

**Figure 3.** Approximate calculation of the region under a curve above the interval [0;1] in the dynamic mathematics system The Geometer's Sketchpad.

Consider another problem—the movement of a point along a straight line. Let us denote the coordinate of the point on the straight line at the time t as X(t); then, the average velocity of the point V(t) for the time from t to the next moment of the timing of the movement t + h can be approximately calculated as $V(t) = \frac{X(t+h) - X(t)}{h}$. From this equality it is possible to build a computational scheme of successive positions of a point on a straight line X(t + h) = X(t) + V(t)· h, for example, for a time interval from 0 to 1. This algorithm is easy to implement dynamically and see the movement of a point on a screen.

**Computational algorithm**

*t := 0;*
*while t < 1*

$X := X + h \cdot V(t);$
$t := t + h$
**end while**

It can be seen that, up to the notation of variables, both computational schemes are the same. The second algorithm does not specify the initial position of the point. If we add $X := 0$ at the beginning, then the algorithms will match completely. The ability to change the initial value of X indicates that the point can start moving from different initial positions and move along different trajectories with the same velocity change function. In terms of antiderivatives, X(t) is called the antiderivative of V(t), and the described result says that the antiderivative of a function V(t) is determined up to a constant value. Knowing that the speed V(t) is the derivative of the coordinate X(t) with respect to time t, we arrive at the idea of connection between the concept of antiderivative and the concept of derivative.

Thus, a comparison of computational schemes for different problems makes it possible to reveal the commonality between various mathematical concepts. The presented case study shows the relationship between the representations "the area of region under a curve" and "the coordinate of a point moving with a given velocity".

### 3.5. Combinatorial Identities and Generating Functions

Trigonometric identities are well represented in the school curriculum, and combinatorial identities are much less so. The former are well developed in the school methodology, while the latter receive much less attention. The reason, in our opinion, is the greater content depth of the latter and the impossibility at the school level to build their study on the basis of operational culture.

Consider two simple combinatorial identities:

$$\sum_{k=0}^{n} \binom{n}{k} = 2^n \text{ and } \sum_{k=1}^{n} \left( k \cdot \binom{n}{k} \right) = n \cdot 2^{n-1}$$

Each of them can be comprehended in two interpretations: combinatorial and algorithmic (Table 1).

**Table 1.** Combinatorial (left) and algebraic (right) interpretations.

| Combinatorial Interpretation | Algebraic Interpretation |
|---|---|
| $\binom{n}{k}$ is the number of k-element subsets of a set of $n$ elements. $\sum_{k=0}^{n} \binom{n}{k}$ is the number of all subsets of a set of $n$ elements. $2^n$ is the number of all subsets of a set of $n$ elements, calculated differently, namely, equating two calculations that lead to the same result. by the product rule, considering that each element has two possibilities—to be chosen or not. $$\sum_{k=0}^{n} \binom{n}{k} = 2^n$$ | $(1+x)^n = (x+1) \cdot (x+1) \cdot ... \cdot (x+1) =$ $1^n + \binom{1}{n} \cdot 1^{n-1} \cdot x + ... + \binom{n}{k} \cdot 1^{n-k} \cdot x^k + ... + x^n$ Binomial theorem. Models isomorphism: the product of $n$ brackets of the form $(x^0 + x^1)$ is considered, each of which can be interpreted as $(x^0 + x^1)$. Choosing an $x$ from a parenthesis corresponds to choosing 0 or 1 for the corresponding member of the binary set. The reduction of like terms will automatically count the number of subsets with the same number of elements. If we substitute 1 instead of $x$, then the sum of the numbers of all subsets will be on the right side. On the left side, we will automatically get $2^n$. |

Thus, the essence of combinatorial calculation is to compare different computational models for counting the number of combinations. The number of combinations can be thought of as the number of loop steps that generate countable combinations. We can represent them with different algorithms:

**Computational algorithm 1**
**do** *over k from 1 to n*
*Construct all subsets from k elements*
**end of do**
**Computational algorithm 2**
*do*

*Construct the next binary set and match it with a subset of elements that correspond to the units of the binary set*

**until** *no more binary sets of n elements*

**end of do**

In fact, different algorithms use different data structures. While the first algorithm constructs the subsets directly, the second encodes the subsets as sets of zeros and ones. Thus, different interpretations can be associated with different data structures.

Indeed, when teaching courses on discrete mathematics, difficulties arise in explaining the complexity of algorithms if they use different data structures.

Algorithmic interpretation partially collapses the calculation process: in the algebraic version, due to the implementation of algorithms for working with polynomials, the process of counting subsets with the same number of elements is encapsulated. Instead of solving one problem, we get a solution to many problems—simultaneously counting the number of all subsets with the same number of elements.

More surprising is that, being within the framework of algebraic interpretation, we can easily obtain the second identity by differentiating the Newton binomial and substituting x = 1:

$$n \cdot (1+x)^{n-1} = \binom{1}{n} \cdot 1 + \binom{2}{n} \cdot 2 \cdot x + ... + \binom{k}{n} \cdot k \cdot x^{k-1} + ... + n \cdot x^{n-1}$$

"Moving backward" we can compare the computational algorithms for generating combinatorial objects for the left and right sides, but for this we need to find combinatorial objects that allow these computational schemes. If the objects are known—subsets with a distinguished element—then the interpretation of the calculations will not be difficult. The question remains: how was the desired combinatorial object guessed? This is the creative part of the computational problem.

Interestingly, in combinatorial problems, the calculation formulas themselves often give an idea of which combinatorial problem was solved. For example, n! is associated with the calculation of permutations, and the sign of multiplication with a combination of independent features, from which expressions of the form $a^n$ can be interpreted. Division is associated with the idea of factorization, and addition with the division of the set of combinations into subsets of objects for which the number of combinations can be counted using known formulas.

The program Wise Tasks Combinatorics [8] is built on the idea of a connection between algebraic and computational interpretations.

In this system, the problem is described by a program that generates all combinations that are obtained from simple sets through their Cartesian products, unions, and other binary operations on sets that are used in describing combinatorial problems. The program goes through all such combinations and counts their number. The interface provides students with the ability to enter arbitrary arithmetic expressions, supplemented by such combinatorial functions as the factorial and the number of k-element subsets of a set of n elements.

All calculations with expressions entered by a student are performed on the set of rational numbers, for which the number of digits of the numerator and denominator is not limited (long numbers).

The result is compared with the result calculated by the program and reported to the student (Figure 4).

Thus, the student's answer is compared not with the teacher's answer, but with the answer that is generated automatically according to the condition of the problem. If the compiler of the problem makes an error in the condition, this leads to the fact that another problem is actually formed, and the system will check the solution of this particular problem. Tasks can be posed by a student who would look for answers, and the system will check the correctness of the answer of any task allowed by the system. For example, Figure 4 shows three different expressions that define the same answer.

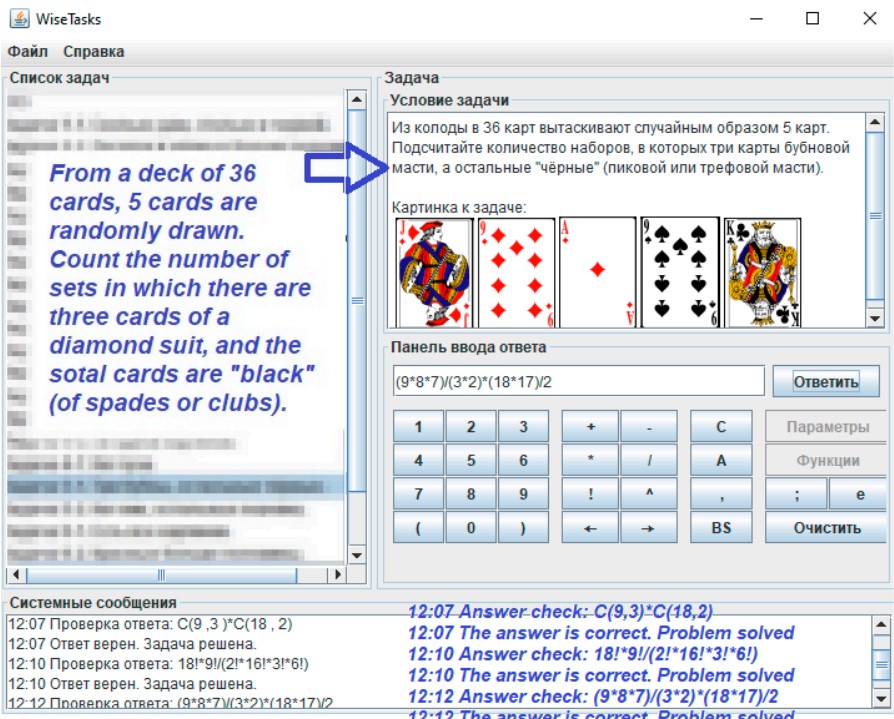

**Figure 4.** Wise Tasks Combinatorics system. The system allows us to check tasks by their description, and it does not matter which formula represents the answer.

Therefore, the data structure of the task, which allows its convenient description, allows one to set and check tasks.

The only difficulty for the compiler of tasks is the need to use a special xml-language for describing the conditions of tasks. The authors of the system [8] found the following solution: instead of using a common editor for composing tasks, thematic editors were created that allow one to set tasks by changing the parameters of task conditions. For example, there exist the editor of tasks on maps, the editor of tasks on numbers, the editor of tasks on words, the editor of tasks for coloring polygons and polyhedra, the editor of tasks on a chessboard, etc.

## 4. Discussion

In the considered case studies, the role of calculations is different: from filling in tables, which are protocols for the execution of algorithms, to a comparative analysis of the algorithms themselves. Important for this work is the question of the transition from calculations by hand to computer ones. When making calculations by hand, the student performs two roles: the organizer and the executor of the calculations. Working with a computer, the student retains only the role of the organizer, outsourcing the execution of calculations to the computer. In such a setting, the following risks can be distinguished:

(1) The performance of elementary computational actions by a student can be considered as training of elementary mental mechanisms, the failure of which may have delayed consequences, that can only be assessed in a longitudinal study with the participation of psychologists;

(2) When outsourcing computations to a computer, a person must be sure of the correctness of their implementation.

In our theoretical analysis, only the second question can be answered. This answer is presented in methodically developed case studies related to the introduction of new concepts. In some case studies, computational schemes were used, in which students partially performed calculations "by hand", while in others, the implementation of computational

schemes was completely carried out on a computer. Let us analyze the role of computations in different case studies.

In the case study "Positional numeral systems and information compression algorithms", a connection was built between explaining the ideas of the decimal number system to younger students based on actions with sets of objects and the algorithm for moving to a new base. It is shown how the comparison of calculations of string lengths in unary and decimal systems can serve as a basis for introducing the concept of information compression. Thus, in this case study, calculations were used so that schoolchildren independently obtained experimental data for comparison and felt the difference in the growth of records of the same number in two different coding systems.

The case study of "*p*-adic numbers and the algorithm "division with remainder"" showed that a simple division algorithm with a remainder can become the basis of theoretical generalizations and allow one to come to an understanding of the complex concept of a *p*-adic number in ways accessible to a schoolchild. The encapsulation of this algorithm allows schoolchildren to comprehend the idea of a reverse code, which is used in computer processors, without additional effort. This case study shows that in some situations the computational algorithm reveals the concept better than its formal definition. This case study uses a psychological phenomenon that is well known to mathematics teachers: students who find it difficult to give a definition, but who have the right ideas about a mathematical concept, instead of giving a formal definition, offer to show how a particular concept works in an algorithm and provide calculations illustrating this concept.

The case study "Diophantine equations, continued fractions and Euclidean algorithm for finding GCD" presents a computational scheme implemented in the form of a computer module, but requiring calculations by hand to solve various problems. It is shown how one computational scheme can serve as a basis for a general look at such different concepts as a continued fraction, a Diophantine equation, and a reciprocal number in modular arithmetic. The computational scheme has become here a means of "enlarging didactic units" [9], allowing one to see what is common in different mathematical concepts and make the computational scheme the basis for theoretical reasoning. This case study shows that computational schemes can become a means of interiorization (and subsequent encapsulation) of concepts: filling in computational protocols and comparing them with each other leads to the generalizations that the teacher plans.

In the case study "Exponential function and Euler's computational scheme", by discretizing the algorithm for solving a simple differential equation, a connection was made between the definition of an exponential function and a geometric progression, and thus with the definition of the y = ax function, which is based on a generalization of the idea of repeated multiplication of a number by itself. This case study shows that the consideration of computational schemes for simple discrete models makes it possible to connect the main ideas of calculus with the ideas of algebra and number theory.

In the case study "The concept of the integral and the approximate calculation of the derivative", as in the case study "Diophantine equations, continued fractions and the Euclidean algorithm", one computational scheme describes the solution of different problems and thus it becomes a mechanism for generalizing and forming the concept of integral and antiderivative. Unlike the case study mentioned above, calculations by hand are not assumed here, but the solution is built in a dynamic mathematics system (spreadsheets can also be used) using any software environment that allows one to visualize the movement of a point (Scratch, Python, or JavaScript). This case study shows the importance of connecting the ideas of computer science and mathematics in a student's rich computer environment. In computer-free learning, it is actually assumed that the definition of the integral as the limit of finite sums is already encapsulated in the student's intellectual mechanisms. In fact, it turns out that few students can use this definition in problem solving. In accordance with the works of Vygotsky [10], Leontiev [1], Papert [11], and Dubinsky [3], in order for these actions to be encapsulated into concepts, they must be brought outside, and then the student's actions with them in the external environment

will lead to their internalization into internal processes, which are then encapsulated into concepts. Dubinsky calls this approach de-encapsulation of mathematical concepts [3]. In these terms, we can say that most of the case studies discussed in this paper demonstrate the de-encapsulation of various mathematical concepts.

In the case study "Combinatorial identities and generating functions", it is shown that computational algorithms for enumerating combinatorial objects can become the basis for the formation of combinatorial thinking. On the other hand, the ability to accurately describe a set of combinatorial objects provides the basis for creating a new type of tasks (Wise Tasks) [9,12], which have the property of checking the correctness of the answer according to the description of the condition, and not according to the reference answer. Also, in this case, the features of using symbolic algebra systems such as Mathematica, Maple, etc. for the algebraic solution of combinatorial problems were discussed. Important here is the transition from one interpretation to another and back. In this case, concepts encapsulated in one computational scheme can be de-encapsulated in another, which is the basis for understanding [3].

The analysis does not present mechanisms for conceptualizing computation that go beyond already known theories. So, for example, in work [13] it is shown that the search for a Turing machine with a fixed number of states and a binary alphabet, outputting the longest result to the tape and stopping, distinguishes between human and computer solutions (brute force algorithm). A person, using his/her existing conceptual knowledge, obtains twice as bad a result as a brute-force algorithm. At the same time, a person finds it difficult to justify the solution proposed by the computer. This problem shows a further direction of research: the study of the mechanisms of constructing concepts within the framework of which it is possible to explain the solutions found by the computer.

## 5. Conclusions

What conclusions can be drawn from the conducted theoretical and methodological analysis?

1.  Reading the work of I.S. Shapiro, written more than 40 years ago, shows that in modern conditions it is impossible to expect the action of psychological mechanisms that are formed during the algebraic transformation of trigonometric expressions. The appearance in the environment of computing tools that duplicate calculations that the student traditionally performed by hand raises the problem of preserving, under new conditions, the psychological effect that forms the student's intellectual mechanisms (convolution of algorithms, and encapsulation of algorithms into mathematical concepts), which was previously achieved by "manual" calculations. It is necessary to clarify the implementation of the ideas of the activity approach to learning, when the performer of operations is not a student, but a computing device. Solving this problem requires serious psychological longitudinal research.
2.  The students' knowledge of algorithms related to mathematical concepts can often be identified with the students' subjective feelings of understanding of these concepts. Therefore, the implementation of algorithms according to transparent computational schemes contributes to overcoming formalism in the study of mathematics, as it forms the feeling in schoolchildren that they themselves can engage in mathematical activities.
3.  The use of various environments that execute mathematical algorithms implies the possibility of using these algorithms in a "logically encapsulated" form. In order to achieve "psychological encapsulation", it is required to de-encapsulate the algorithm, that is, to deploy it in the form of a computational circuit that is available for a student to check.
4.  Some simple computational schemes, such as Euler's scheme for solving differential equations, can serve as the basis for generalizations that students themselves can make, revealing the commonality of computational schemes for solving problems in different representations of mathematical concepts.
5.  The presence of "mathematical solvers" gives more weight to the ability to correctly set problems. Connecting different computation schemes, such as a naive enumeration

scheme with an efficient one, provides a framework to support research activities in which the student's intelligence interacts with "artificial intelligence" (AI) in solving a problem (AI in this situation is represented by powerful calculators based on a "force" solution of the problem, in which the lack of mathematical theory and effective algorithms is compensated by a simple enumeration of options).

6. The introduction of mathematical concepts through computational processes requires the attention of methodologists to the data structures used in computations. Different interpretations of the same problem can generate different computational schemes due to the different data structures to which the algorithms are applied. The importance of studying data structures in the study of mathematics and computer science has not yet received due attention, although the practice of introducing schoolchildren to the concept of the complexity of algorithms is becoming increasingly common.

**Author Contributions:** Conceptualization, E.T. and A.L.; methodology, A.L.; validation, E.T., A.L. and S.P.; formal analysis, A.L.; investigation, A.L.; resources, S.P.; writing—original draft preparation, S.P.; writing—review and editing, S.P.; visualization, S.P.; supervision, S.P.; project administration, S.P. All authors have read and agreed to the published version of the manuscript.

**Funding:** This research received no external funding.

**Data Availability Statement:** Not applicable.

**Conflicts of Interest:** The authors declare no conflict of interest.

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
