# Peer review of "Computational “Accompaniment” of the Introduction of New Mathematical Concepts"

_computation, doi:10.3390/computation11100194_

Round 1
Reviewer 1 Report
Many thanks for this paper. You outline a workable, I think, approach to the key issue of the mathematics education of our times: the balance between the conceptual understanding of mathematics and the use of computer-assisted mathematics. Moreover, you one of the first people who clearly understand this issue. Discussion of this issue needs to be continued, I and I hope to see your further work in this direction.
Author Response
Thank you very much for your interest in the article.

Reviewer 2 Report
In general, the authors have made considerable efforts to study ICT and computational tools used in the introduction of mathematical concepts. Computational modules claim a new approach supported by psychology in the absence of conceptual control. Any new contribution to this topic that offers new insights into the new psychology of conceptual control with respect to an optimal experience using different computational tools and AI is greatly appreciated at this point.
To further strengthen the manuscript, here are some thoughts on this topic:
- References should be cited as they appear in the body of the text...the …first reference (Leontiev) should be [1], followed by [2] (Shapiro)…- Align > both in the body of the text and in the list of references….
- In Chapter 4. discussion: some limitations of this study should be added, along with future work…where a use of GPT4-based or other mathematical applications and tools could be interesting to contrast and evaluate current knowledge and psychology….
- The formatting of the references needs to be adapted to the journal template….
The English language is sufficiently intelligible and well structured in the manuscript. The lexical sophistication is satisfactory, the text is well structured, and the syntactic forms and variety of syntactic forms are also sophisticated. The interconnectivity of text segments based on text features is present and satisfactorily indicates lexical, semantic, and argument dependencies within a text. All in all, the text is easy to read and the structure of the text well reflects the intended topic.
Author Response
Thank you very much for your interest in the article.
In accordance with your comments, corrections have been made to the list of references.
Also added are restrictions on the results presented in the articles and prospects for further work, taking into account the rapid development of artificial intelligence.
Below is the added fragment, and in the appendix is the new edition of the article.
"The analysis does not present mechanisms for conceptualizing computation that go beyond already known theories. So, for example, in the work [13] it is shown that the search for a Turing machine with a fixed number of states and a binary alphabet, outputting the longest result to the tape and stopping, distinguishes between human and computer solutions (brute force algorithm). A person, using his existing conceptual knowledge, gets twice as bad a result as a brute-force algorithm. At the same time, a person finds it difficult to justify the solution proposed by the computer. This problem shows a further direction of research: the study of the mechanisms of constructing concepts within the framework of which it is possible to explain the solutions found by the computer."
